# The calmodulin redox sensor controls myogenesis

**Alex W. Steil, Jacob W. Kailing, Cade J. Armstrong, Daniel G. Walgenbach, Jennifer C. Klein** [ID] *

Department of Biology, University of Wisconsin-La Crosse, La Crosse, WI, United States of America

* jklein@uwlax.edu

## Abstract

Muscle aging is accompanied by blunted muscle regeneration in response to injury and disuse. Oxidative stress likely underlies this diminished response, but muscle redox sensors that act in regeneration have not yet been characterized. Calmodulin contains multiple redox sensitive methionines whose oxidation alters the regulation of numerous cellular targets. We have used the CRISPR-Cas9 system to introduce a single amino acid substitution M109Q that mimics oxidation of methionine to methionine sulfoxide in one or both alleles of the *CALM1* gene, one of three genes encoding the muscle regulatory protein calmodulin, in C2C12 mouse myoblasts. When signaled to undergo myogenesis, mutated myoblasts failed to differentiate into myotubes. Although early myogenic regulatory factors were present, cells with the *CALM1* M109Q mutation in one or both alleles were unable to withdraw from the cell cycle and failed to express late myogenic factors. We have shown that a single oxidative modification to a redox-sensitive muscle regulatory protein can halt myogenesis, suggesting a molecular target for mitigating the impact of oxidative stress in age-related muscle degeneration.

**Data Availability Statement:** All relevant data are within the paper and its Supporting Information files.

**Funding:** This work was supported by the National Institutes for Health (Klein R15AG048286). AWS,

## Introduction

Aging causes a decline in muscle power and speed due to muscle atrophy and weakness; by the age of 80, humans lose 40% of muscle mass and 50% of muscle power [1]. The primary alteration in aged muscle is a disproportionate decrease in protein synthesis, increase in protein breakdown [2], and functional dysregulation of muscle proteins [3–6]. Exercise is a potent modulator of muscle physiology and can prevent some, but not all, of the negative impacts of aging on muscle strength [2, 7]. In healthy adults, injured or overused muscle is regenerated through muscle satellite cells. These are stem cells in the myoblast stage of differentiation that can differentiate into myotubes, which fuse with existing fibers to repair damage. A predominant feature of muscle aging is the diminished ability to regenerate muscle fibers after injury or wasting associated with cancer, congestive heart failure, and broken bones [8, 9]. Not only are there fewer satellite cells in aged muscle, their capacity for proliferation, activation, and differentiation is reduced [9]. Aged muscle satellite cells experience higher levels of oxidative stress, reduced antioxidant capacity, disrupted protein homeostasis, epigenetic alterations, and

DGW, and JWK were supported by University of Wisconsin La Crosse Undergraduate Research and Creativity Awards. The funders had no role in study design, data collection and analysis, decision to publish, or preparation of the manuscript.

**Competing interests:** The authors have declared that no competing interests exist.

reduced mitochondrial activity [10]. Oxidative stress is known to adversely influence myoblast survival and ability to repair damaged muscle [11]. Indeed, in vitro myogenesis is impaired by oxidative stress and supported by a reducing environment [12–16]. Although it is clear that signaling via reactive oxygen species (ROS) regulates myogenesis at numerous points [17], it is imperative to identify specific protein redox sensors that are targets of cellular oxidation and serve to propagate redox signals throughout metabolic and transcriptional networks, orchestrating changes in muscle physiology that ultimately define human healthspan.

Proteomics approaches have been useful in identifying muscle regulatory and contractile proteins that are oxidatively modified with aging [18, 19]. Systematic biochemical approaches by our group and others have shown that these oxidative modifications impact muscle protein structure and function [20–23]. Two decades of work suggest that the muscle regulatory protein calmodulin (CaM) is poised to act as a key redox-sensitive modulator of muscle physiology. CaM is the brilliant conductor of the cellular orchestra that plays in response to intracellular $Ca^{2+}$, playing pivotal roles in $Ca^{2+}$ dynamics and signaling, muscle contraction and remodeling, metabolism, autophagy, cell proliferation, and cell death [24–26]. There is evidence that CaM modulates its response to $Ca^{2+}$ according to cellular redox status. CaM has unusually high methionine (Met) content, including 46% of the hydrophobic residues in its binding pockets, which are crucial for CaM binding to hundreds of diverse targets. CaM's nine Met are susceptible and functionally sensitive to cellular oxidants [27–29]. CaM isolated from aged brains cannot fully activate the plasma membrane $Ca^{2+}$ ATPase due to age-related Met oxidation that blocks productive association between CaM and the $Ca^{2+}$ ATPase [30, 31]. Progressively oxidized CaM (in vitro) cannot fully activate CaMKII [32, 33], adenylyl cyclase [34], or nitric oxide synthase [35]. CaM's C-terminal Mets, particularly Met 109 and Met 124, are crucial for ryanodine receptor (RyR) channel regulation [36, 37]. Our group has delineated the molecular structural mechanism by which oxidation of CaM's C-terminal Met (M109 and M124, see Fig 1B) trigger changes in CaM conformational dynamics [20, 38]. Thus far,

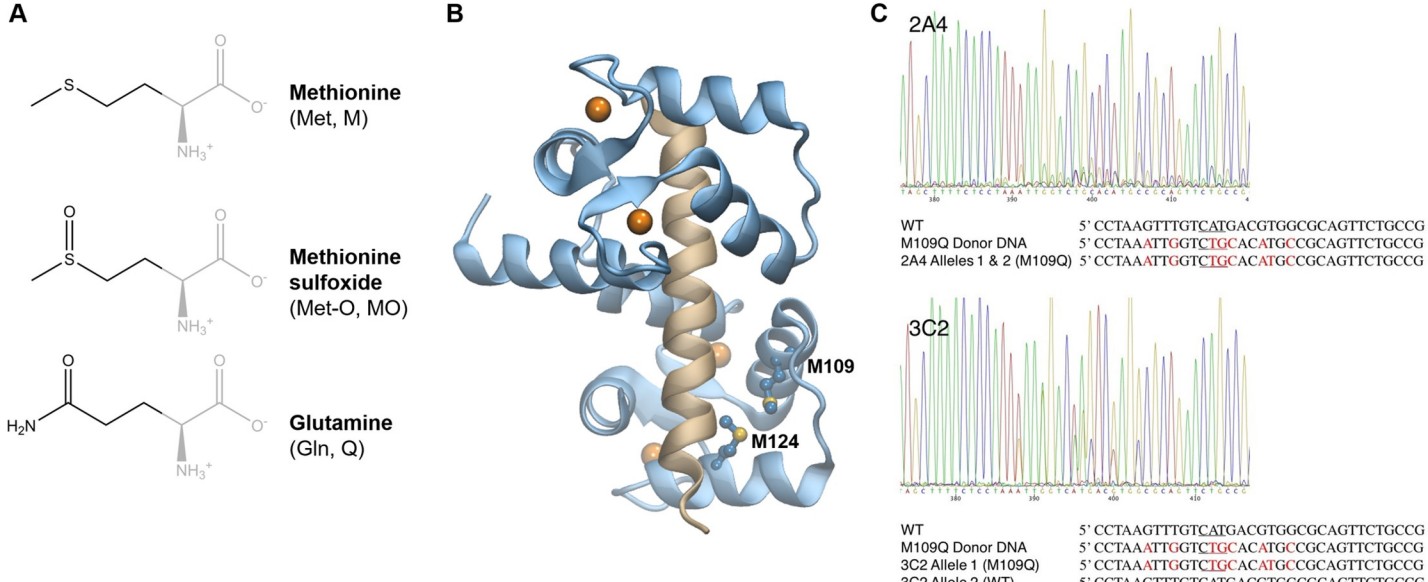

**Fig 1. *CALM1* gene editing in mouse myoblasts.** (A) Methionine can be oxidized to methionine sulfoxide by cellular oxidants; glutamine substitution mimics methionine sulfoxide. (B) Structural models of CaM (blue) bound to the RyR peptide (tan) with four bound $Ca^{2+}$ atoms (orange); CaM Met 109 and Met 124 sidechains are shown (PDB: 1BCX). (C) CRISPR-Cas 9 editing of C2C12 myoblasts. Sanger sequencing of the target region in *CALM1* for the 2A4 and 3C2 cell lines. The wild type (WT) sequence, donor DNA sequence, and the mutant sequences are shown (antisense). Mutations are indicated in red. Nucleotides encoding M109(Q) are underlined in red.

biochemical investigations of CaM-target protein interactions and biophysical investigations of CaM conformational dynamics indicate that CaM is poised to act as a potent cellular redox sensor. However, nobody has characterized the site-specific impact of CaM Met oxidation on muscle physiology.

Muscle regeneration occurs when muscle satellite cells undergo myogenesis, a precisely orchestrated process that includes (1) the activation of muscle-specific transcription factors, (2) phenotypic differentiation that includes the expression of muscle contractile and regulatory proteins, (3) and fusion of muscle cells to form a mature, multinucleated myotube. Myogenic regulatory factors (MRFs) including MyoD, Myf5, myogenin, and MRF4, are basic helix-loop-helix (bHLH) transcription factors that direct progenitor cells to commit to the muscle cell lineage and carry out the differentiation program [39]. MyoD and Myf5 are expressed in undifferentiated, proliferating myoblasts, while myogenin and MRF4 are induced after cell cycle withdrawal during later phases of differentiation [40, 41]. Myoblast commitment to the differentiation pathway is marked by myogenin expression. The myogenin promoter is regulated via two distinct enhancer regions, an A/T rich element recognized by MEF2 transcription factors and an E-box recognized by MyoD [42]. MyoD is involved not only in myoblast proliferation and the transcriptional cascade that promotes muscle cell differentiation, but paradoxically, also triggers cell cycle arrest. MyoD drives the expression of at least three cell cycle inhibitors that include p21, a cyclin-dependent kinase inhibitor that blocks entry to the cycle cell at S phase [43], retinoblastoma growth suppressor (Rb) [44], and cyclin D3 [45], which together induce terminal cell cycle withdrawal. Once the myoblast has withdrawn from the cell cycle and committed to differentiation, structural and contractile proteins are expressed, and myoblasts fuse with damaged fibers.

$Ca^{2+}$ is a crucial component of the muscle differentiation medium [46], and $Ca^{2+}$ signaling pathways are now implicated at every step of myogenesis [47]. Downstream effectors of $Ca^{2+}$ are numerous, but $Ca^{2+}$-bound CaM (CaCaM), CaCaM-dependent kinases, and the CaCaM-calcineurin complex play key roles in facilitating myogenesis [48]. Given CaM's central role in $Ca^{2+}$ signaling and CaM's structural and functional sensitivity to oxidation, we hypothesize that CaM serves as a muscle redox sensor that blocks myogenesis when the redox balance tips toward oxidative stress. We have used the CRISPR-Cas9 system to knock-in a single amino acid mutation that mimics the oxidative modification of a single Met to Met sulfoxide (M109Q) in one or both alleles of the *CALM1* gene, one of three genes encoding the muscle regulatory protein CaM in mammalian cells. We have shown that a single oxidative modification to a redox-sensitive muscle regulatory protein can halt myogenesis, suggesting a molecular target for mitigating the impact of oxidative stress in age-related muscle degeneration.

## Methods

### C2C12 cell culture and differentiation

C2C12 cells from American Type Cell Collection (Manassas, VA, USA) were cultured in DMEM growth medium (GM) containing 25 mM glucose, 4 mM L-alanyl-glutamine (Gluta-Max), 10% fetal bovine serum (FBS), and 1% penicillin/streptomycin (ThermoFisher, USA) at 37°C and 5% $CO_2$. Cells were passed at 75% confluency using 0.25% trypsin to lift adherent cells; cells were discarded after 20 passages. Differentiation was induced by removing FBS from the growth medium and replacing it with 2% horse serum (ThermoFisher). The addition of differentiation medium (DM) is defined as day 1 of differentiation. Cells were typically differentiated for 96 hours, with medium replacement every 48 hours.

## CRISPR gene editing, clonal isolation, and sequencing

The Alt-R CRISPR-Cas 9 system (Integrated DNA Technologies, USA) was used to introduce a point mutation in the mouse *CALM1* gene (Gene ID: 12313). The gRNA was composed of a complex of a crispr RNA (crRNA) that directs Cas9 to the cut site and a transactivating RNA (tracrRNA) that serves as a scaffold for attaching the crRNA to Cas9. Several crRNAs were assayed for cutting efficiency. A long (60 bp) single-stranded DNA oligo was synthesized by Integrated DNA Technologies to be used as a donor DNA (dDNA) to knock-in mutations near the cut site (Fig 1C). The gRNA was made by incubating 5 μM crRNA with an equimolar amount of tracrRNA in nuclease-free duplex buffer at 95°C for 5 mins. The Cas9:gRNA ribonucleoprotein (RNP) complex was made by incubating 1 μg/μL Cas9, 5 μM (170 ng/μL) gRNA, and Cas9 PLUS reagent in OptiMem (ThermoFisher) for 5 mins at 25°C. RNP complexes were prepared for transfection by incubating the RNP complex and dDNA in CRISPR-MAX lipofectamine (ThermoFisher) for 15 mins at 25°C. Cells were transfected by seeding $10^5$ cells in DMEM with 10% FBS, but no antibiotics, followed by immediate addition of the prepared lipofectamine/RNP/dDNA mix while cells were still suspended. Cells grew under normal conditions for 24 h before they were harvested for both clonal isolation and a genomic cleavage detection assay (Integrated DNA Technologies) to estimate cutting efficiency.

**Genomic cleavage detection assay.** Genomic DNA was prepared from CRISPR-treated C2C12 cells using QuickExtract (Lucigen, Madison, WI, USA). The targeted region of *CALM1* (400 bp) was PCR amplified using AmpliTaq Gold (ThermoFisher). The PCR amplicons were heated to 95°C and then slowly re-annealed so that mismatches would occur between native (unmutated DNA) and CRISPR-edited DNA. Mismatches were detected using T7 endonuclease (New England Biolabs, Ipswich, MA, USA), which cuts mismatched regions of duplex DNA, including single base mismatches. The extent of T7 cutting was quantified using gel electrophoresis.

**Screening for CRISPR-edited cells.** CRISPR-edited cells were diluted to 12 cells/mL and used to seed a 96-well plate to isolate individual transfected cells. Colonies which grew from isolated cells, identified by phase contrast microscopy, were expanded for 2–3 weeks. Genomic DNA was prepared from approximately 200 expanded clonal isolates using QuickExtract (Lucigen). The targeted region of *CALM1* (400 bp) was PCR amplified and treated with BspHI (New England Biolabs), which cut native dsDNA, but not CRISPR-edited dsDNA. DNA from approximately 40 samples that passed the screen were sequenced via Sanger sequencing (Eton Bioscience, USA).

## Immunofluorescence labeling and microscopy

Cells were differentiated in 96-well plates and then fixed at time-points with 5% formaldehyde (ThermoFisher) for 15 min, permeabilized with 0.5% Triton-X 100 (ThermoFisher) for 15 min, and blocked with 5% w/v BSA in PBS for 1 h. Primary antibodies include anti-MyoD G-1 (1:250, Santa Cruz Biotechnology, USA), anti-p21 F-5 (1:50, Santa Cruz Biotechnology), anti-p53 DO-1 (1:50, Santa Cruz Biotechnology), anti-phospho-p53 Ser 315 (1:50, Santa Cruz Biotechnology), anti-myogenin F5D (1:250, ThermoFisher), and anti-MHC MF20 (1:50, R&D Systems, USA). All primary antibodies were diluted in 5% w/v BSA in PBS and applied to blocked myotubes for 1 h. All primary antibodies were visualized using 1:200 Alexa Fluor 488 goat anti-mouse IgG (ThermoFisher). Nuclei were stained using NucBlue Live Cell Stain (ThermoFisher) for 15 min, and samples were preserved using Prolong Gold Antifade (ThermoFisher). All experiments were performed in triplicate. Images were acquired using an Eclipse Ts2R inverted fluorescence microscope (Nikon, Melville, NY, USA), a Zyla sCMOS camera (Andor, Belfast, UK), and NIS Elements basic research software (Nikon).

## Live cell imaging

Cells were imaged in Live Cell Imaging Solution (ThermoFisher) in cell culture flasks or plates. MitoTracker Red and LysoTracker Deep Red (ThermoFisher) were used as recommended by the manufacturer.

## Image analysis

CellProfiler [49] was used to detect and quantify nuclei and myotubes. Myogenin abundance was quantified as a ratio of nuclei expressing myogenin to total nuclei. Myotube abundance was quantified as a ratio of the number of nuclei in multinucleated myotubes to total nuclei [50].

## Cell proliferation assay

Cells were seeded in a 96-well plate at a density of $10^4$ cells in 100 µL of media. At each time-point, 20 µL CellTiter 96 Aqueous One Solution (Promega, Madison, WI, USA) was added to wells, incubated at 37˚C and 5% $CO_2$ for 2 h, and then absorbance at 490 nm was measured using a SpectroMax 190 plate reader (Molecular Devices, San Jose, CA, USA). Triplicate wells were read for each time-point.

## RT-qPCR

Total RNA was isolated from differentiated C2C12 cells at various time-points using the Pure-Link RNA Mini Kit (ThermoFisher). Reverse transcription of RNA was performed with iScript (Bio-Rad, Hercules, CA, USA) using 1 µg of RNA. The following mouse transcripts were analyzed: beta actin (Actb, Gene ID 11461); cyclin-dependent kinase inhibitor 1A (p21, Gene ID 12575); retinoblastoma tumor suppressor (*Rb1*, Gene ID 19645); myogenin (MyoG, Gene ID 17928); myogenic differentiation 1 (MyoD, GeneID 17927); myocyte enhancer factor 2C (MEF2C, Gene ID 17260); myogenic factor 6 (Myf6, MRF4 Gene ID 17878); mouse calmodulin 1 (CALM1, GeneID 12313; mouse calmodulin 2 (CALM2, GeneID 12314); mouse calmodulin 3 (CALM3, GeneID 12315); inhibitor of DNA-binding 3 (Id3, Gene ID 15903); inhibitor of DNA-binding 2 (Id2, Gene ID 15902; and early growth response 1 (Egr-1, Gene ID: 13653). Beta-actin was used as a reference gene. Real-time PCR was performed using CFX96 Touch Real-Time PCR Detection System (Bio-Rad) and the SsoAdvanced Universal SYBR Green Supermix (Bio-Rad). Reactions were performed using 50 ng of cDNA and included a preincubation (95˚C for 5 min), 45 amplification cycles, and melting curve analysis for verification of specific product.

## Western blotting

**Lysate preparation.** Cell lysates were prepared by resuspending $5 \times 10^6$ pelleted cells in 1X RIPA buffer (ThermoFisher) containing 5X Halt Protease Inhibitor (ThermoFisher) on ice for 15 min, followed by centrifugation at 14,000 x g for 10 min to clear the lysate. Protein concentration was quantified using the BCA Protein Quantification Kit (ThermoFisher). Samples were prepared by mixing 100 µg of lysate with 10 µL 2X Laemmli buffer (Sigma-Aldrich, USA) containing 1 µL beta-mercaptoethanol (Sigma-Aldrich). After adding Milli-Q water to 20 µL, samples were incubated at 95˚C for 5 min.

**SDS-PAGE and transfer.** SDS-PAGE was performed using Mini-PROTEAN TGX Precast Gels (Bio-Rad) and the Mini-PROTEAN Tetra Cell system (Bio-Rad) at 90V. Transfer of protein from the gel to a PVDF membrane were performed using a Trans-Blot Turbo Transfer System (Bio-Rad), a PowerPac Basic (Bio-Rad) power supply, and a Trans-Blot Turbo RTA

Transfer Kit (Bio-Rad). Precision Plus Protein Dual Color Standard (Bio-Rad) was used to monitor electrophoresis and to detect transfer to the PVDF membrane. A biotinylated protein ladder (Cell Signaling Technology, Danvers, MA, USA) was detectable during chemilumine-cent imaging.

**Antibodies.**   The PVDF membrane was blocked in 5% w/v nonfat dry milk in 1X TBS and 0.1% Tween 20 (TBST, Bio-Rad) for 1 h prior to antibody incubations. Actin was probed using a pan-actin D18C11 Rabbit mAb (1:1000, Cell Signaling Technology) diluted in 5% w/v BSA and 1X TBST (Bio-Rad), followed by an anti-rabbit IgG HRP-linked Ab (1:2000, Cell Signaling Technology) diluted in 5% w/v nonfat dry milk and 1x TBST (Bio-Rad), both for 1 h at room temperature. MyoD was detected using the same antibody as used for immunofluorescence. The biotinylated ladder was probed using an anti-biotin HRP-linked Ab (1:2000, Cell Signaling Technology) diluted in 5% w/v nonfat dry milk and 1X TBST (Bio-Rad) for 1 hour at 25C. The membrane was washed three times with TBST for 15 min between each antibody incubation.

**Detection.**   HRP-conjugated secondary antibodies were visualized via chemiluminescence using SignalFire ECL Reagent (Cell Signaling Technology) and the ChemiDoc MP gel documentation system (Bio-Rad).

## Results and discussion

### CALM1 gene editing in mouse myoblasts

Given the importance of $Ca^{2+}$ signaling in myogenesis and the central role of CaM in interpreting the $Ca^{2+}$ signal, we hypothesized that the muscle regulatory protein CaM acts as a cellular redox sensor that blocks myogenesis in response to oxidative stress. To test this hypothesis, we used the CRISPR-Cas9 system coupled with a long single-stranded donor DNA [51, 52] to knock-in a biophysically and biochemically well-characterized [20, 38] single amino acid mutation that mimics the oxidative modification of a single Met to Met sulfoxide (M109Q, Fig 1A and 1B). This was done in one or both alleles of the *CALM1* gene, one of three genes that encodes the muscle regulatory protein CaM in mouse C2C12 myoblasts. C2C12 myoblasts are a standard model for muscle myogenesis; the mononuclear myoblasts undergo myogenesis into multinucleated myotubes within days of mitogen withdrawal from the growth medium [53].

We tested *the only* three potential CRISPR cut sites within 50 bp of M109 for cutting efficiency and we found that one, corresponding to crRNA 5′ CGCCACGTCATGACAAACTT 3′ reliably cut with greater than 50% efficiency. There are no predicted off-target cut sites (< 3 mismatches) for this crRNA in the mouse genome. Myoblasts at a low passage number (< 3 passages) were transfected with gRNA-dDNA-Cas9 complexes. After 24 h, individual cells were isolated and grown to confluency. A restriction assay was used to screen for clonal isolates with mutations near the cut site. About 20% of the 200 clonal isolates screened contained cut site alterations and were sequenced. Of these, only three clonal isolates contained the desired mutations, which included the M109Q mutation and four silent mutations purposefully introduced near the cut site to prevent re-cutting, and no other mutations. The cell line 2A4 contained the desired mutations in both alleles, and the 3C2 and 3C3 cell lines contained the desired mutations in only one allele (Fig 1C). We did not pursue whole genome sequencing to confirm the absence of unwanted mutations because there were no predicted off-target cutting sites and the probability that three independent cell lines would contain both the desired mutation and the same off-target mutation would be < 0.01% given an optimistic cutting efficiency of 20%. All three cell lines produced identical phenotypes (described below), indicating that the mutation is dominantly expressed.

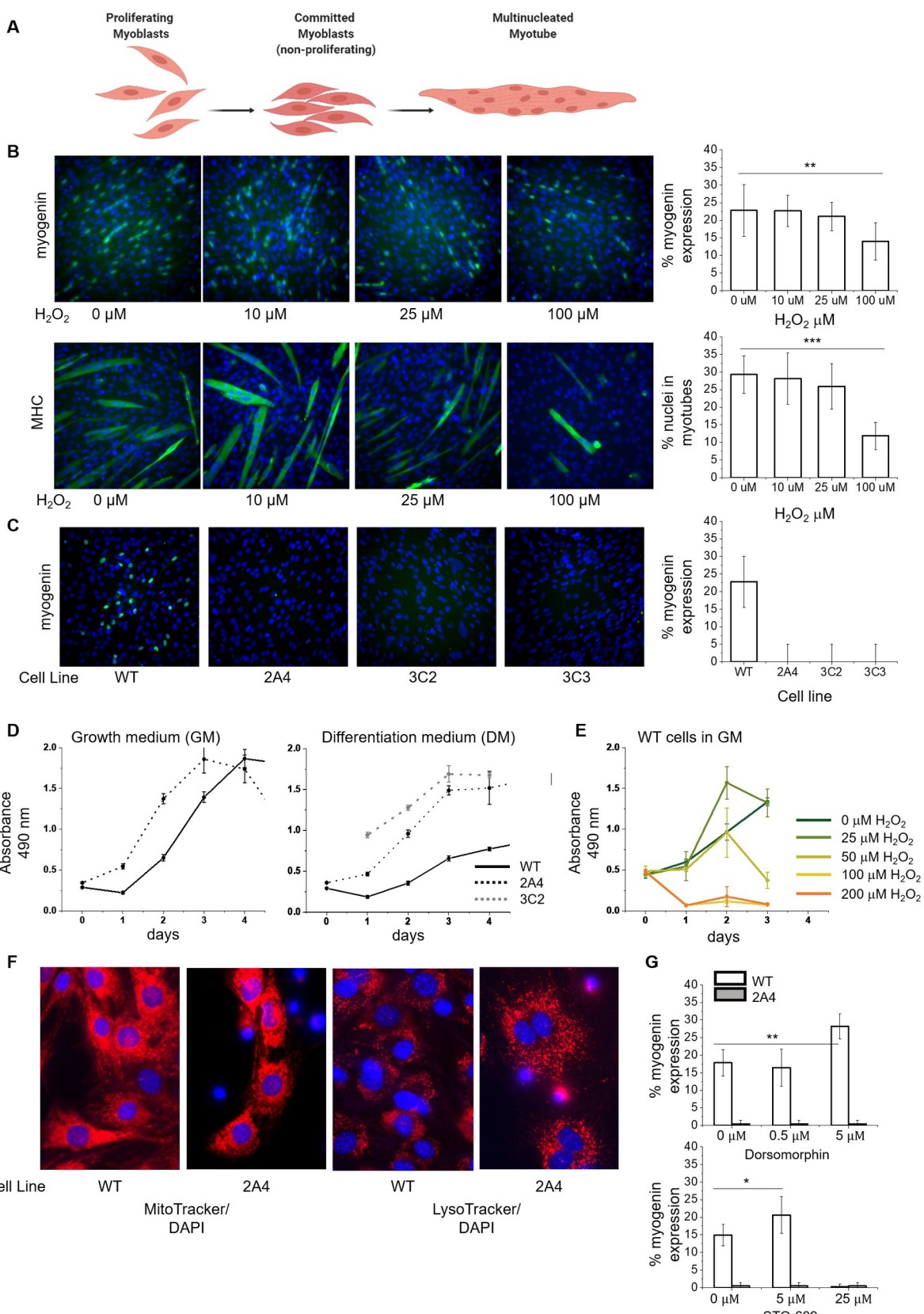

**Fig 2. CRISPR-edited myoblasts fail to exit the cell cycle or produce myotubes.** (A) Proliferating myoblasts are signaled to initiate differentiation by serum withdrawal, followed by myogenin expression, marking commitment to differentiation and permanent cell cycle exit. Muscle-specific proteins are expressed, and myoblasts fuse into multinucleated myotubes. (B) Wild type (WT) C2C12 cells on day 4 of differentiation with chronic exposure to $H_2O_2$ in the differentiation medium (DM). Percentage of nuclei expressing myogenin and the percentage of nuclei in multinucleated myotubes were calculated using Cell Profiler [49] (n = 4). Data are represented as the mean ± SD; **P < 0.01; ***P < 0.001. (C) WT and CRISPR-edited C2C12 cells lines 2A4, 3C2, and 3C3 containing the *CALM1* M109Q mutation in one (3C2, 3C3) or both alleles (2A4) on day 4 of differentiation (n = 4). No myotubes were detected. (D) Proliferation assays for WT, 2A4, and 3C2 myoblasts in GM or DM (n = 2). (E) Proliferation assays for WT myoblasts in GM with $H_2O_2$ (n = 2). (F) Live cell imaging of WT and 2A4 myoblasts stained with MitoTracker Red or LysoTracker Deep Red and DAPI. (G) Percentage of nuclei expressing myogenin in WT and 2A4 myoblasts treated with dorsomophin or STO-609 at the indicated concentrations in the DM (n = 1). STO-609 was toxic to cells at 25 μM. Data are represented as the mean ± SD; *P < 0.1, **P < 0.01.

## The calmodulin redox sensor halts myogenesis

We treated myoblasts with $H_2O_2$ concentrations over which CaM and other targets are expected to be significantly oxidized. When rat ventricular myocytes were exposed to 50 μM $H_2O_2$, there was a 50% reduction in free thiols and a 50% decrease in CaM binding to RyR, an effect attributed to oxidation of both RyR and CaM; pre-treating CaM with 50 μM $H_2O_2$ before perfusion into myocytes (without additional $H_2O_2$-treatment) produced a 20% decline in RyR binding [54]. We found that when the differentiation medium contained 100 μM $H_2O_2$, myogenin expression decreased from 22% to 13% and myotube formation decreased from 29% to 11% in wild type (WT) myoblasts on day 4 of differentiation relative to untreated cells (Fig 2A and 2B), consistent with previous studies using 25 μM $H_2O_2$ to treat C2C12 cells [13]. Myoblasts with the *CALM1* M109Q mutation, either in one allele (2A4) or both alleles (32C or 3C3), could not form myotubes, nor did they express detectable myogenin or myosin heavy chain by day 4 of differentiation (Fig 2C). *CALM1* M109Q myoblasts had normal fibroblast-like morphology, but proliferated at roughly twice the rate of WT cells in growth medium (GM) (Fig 2D). In differentiation medium (DM), WT cells ceased to proliferate; in contrast, *CALM1* M109Q myoblasts proliferated at the same rate in DM as in GM (Fig 2D), indicating that the cell cycle inhibition that normally precedes myogenin expression and terminal differentiation did not occur for *CALM1* M109Q myoblasts. We found that 25 μM $H_2O_2$ enhanced cell proliferation for intermediate time points, but as the $H_2O_2$ concentration approached 100 μM, cells stopped proliferating entirely (Fig 2E), consistent with observations that $H_2O_2$ can enhance proliferation at low concentrations, arrest growth at high concentrations, and elicit apoptosis at even higher concentrations [55].

We investigated mitochondrial structure and lysosome distribution in *CALM1* M109Q myoblasts with live-cell imaging using MitoTracker and LyosTracker to detect redox-regulated alterations to metabolism or autophagy, but didn't observe any differences between WT and 2A4 cells (Fig 2F), even with glucose starvation (not shown). The AMPK nutrient-sensing pathway, which involves $Ca^{2+}$ release and multiple CaM-dependent targets, negatively regulates myogenesis [56, 57], so we hypothesized that inhibitors of this pathway might recover myogenesis in *CALM1*-M109Q myoblasts. We found that although the AMPK inhibitor dorsomorphin and the CaMKKβ inhibitor STO-609 both boosted myogenesis in WT myoblasts, they failed to recover myogenesis in 2A4 cells (Fig 2G).

Immunofluorescence microscopy was used to detect and localize key myogenic regulatory factors in WT and *CALM1* M109Q cells (2A4) undergoing myogenesis. Myosin heavy chain is a marker of phenotypic differentiation, and was expressed in multinucleated WT myotubes by day 2 of differentiation, but was never expressed in 2A4 cells, which failed to produce any multinucleated cells (Fig 3A). Myogenin, the transcription factor that drives phenotypic differentiation, was also present in the nuclei of WT myotubes by day 2 of differentiation, but was never detectable in 2A4 cells. MyoD is a transcription factor that drives exit from the cell cycle at early stages of differentiation, and induces myogenin expression at later stages of

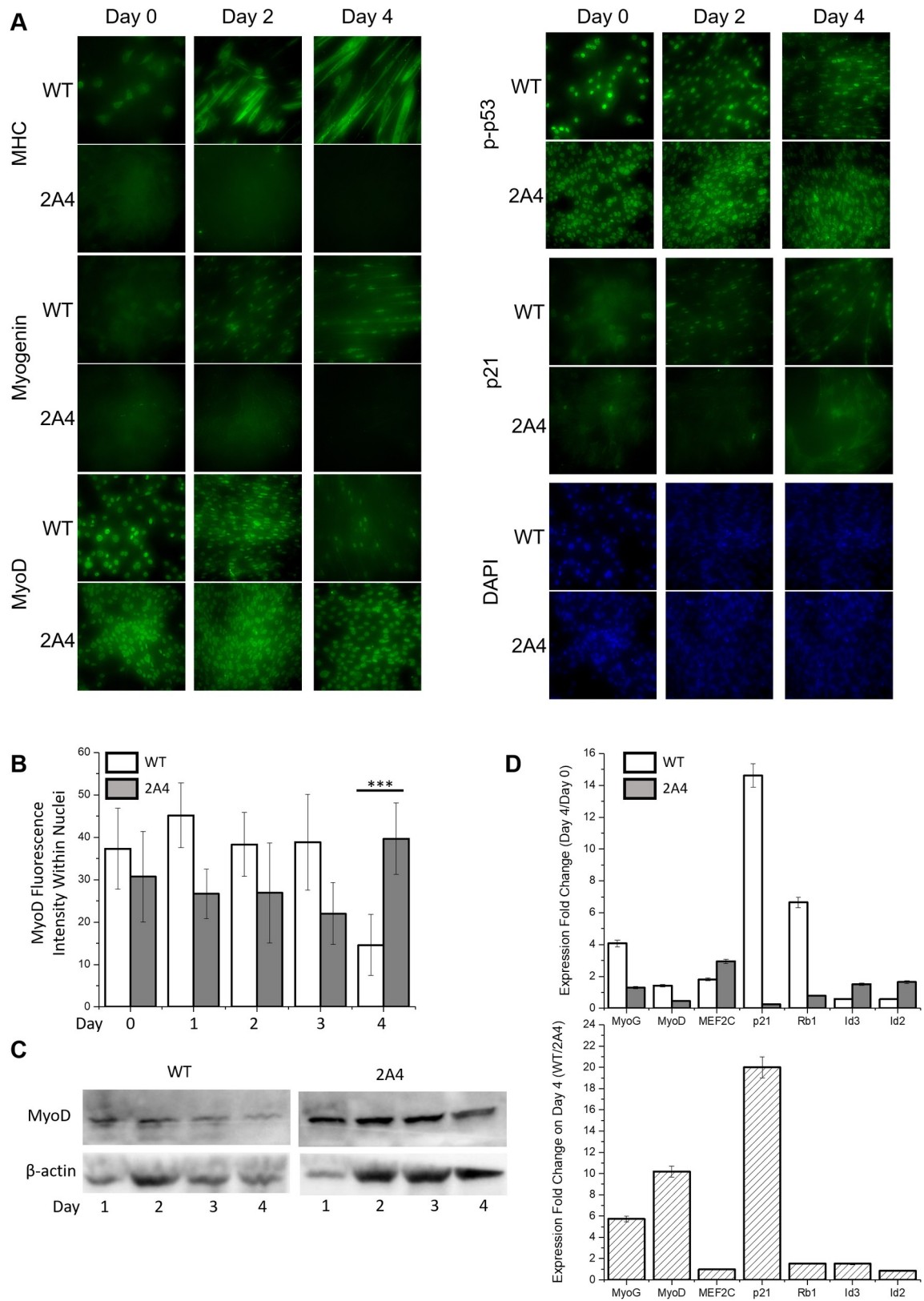

**Fig 3. Expression and localization of myogenic regulatory factors in wild type and CRISPR-edited C2C12 myoblasts during myogenesis. (A)** Immunofluorescence to determine the localization of MyoD, p-p53, p21, myogenin, and MHC in differentiating

myoblasts. **(B)** Average fluorescence intensity of MyoD within nuclei of WT and 2A4 cells on days 0 to 4 of differentiation (average of 10 cells per field, n = 2). Data are represented as the mean ± SD; ***P < 0.001. **(C)** Western blot of WT and 2A4 cell lysates on days 1 to 4 of differentiation (n = 1) to measure MyoD protein level. **(D)** Fold-change in gene expression measured by RT-qPCR for the indicated genes, calculated as the ratio of gene expression on day 4 to day 1 for WT and 2A4 cells (top panel) and as the ratio of WT to 2A4 cells on day 4 of differentiation (bottom panel) (n = 3). Data are represented as the mean ± SD.

differentiation. MyoD was strongly localized to the nuclei of WT myoblasts on day 0 and its presence and nuclear localization persisted to day 3 in both myoblasts and myotubes. In 2A4 myoblasts, MyoD was present and strongly localized to the nuclei of myoblasts from days 0–4. Quantification of fluorescence intensity of MyoD in nuclei (Fig 3B) and western blotting (Fig 3C) indicated that MyoD protein disappeared from WT nuclei on day 4, consistent with its degradation by the 26S proteasome [58], while MyoD protein level was highest at day 4 in 2A4 nuclei. Despite high and persistent levels of MyoD in 2A4 cells, myogenin expression and phenotypic differentiation did not occur. The same regulatory signature was observed in a study using 25 μM $H_2O_2$ treatment to mimic oxidative stress during C2C12 cell differentiation, which caused a steep decline in myogenin expression, but no change in the expression of MyoD or other early transcription factors [13]. As with oxidative stress, the *CALM1* M109Q mutation inhibited the later stages of myogenesis.

We surmised that myogenin expression and MyoD activity might depend on exit from the cell cycle, so we examined the localization and expression of cell cycle inhibitors (Fig 3D). The cell cycle inhibitor p21 is transcriptionally regulated by phosphorylated p53 and MyoD, and drives terminal exit from the cell cycle prior to phenotypic differentiation. Phosphorylated p53 was present in nuclei of both WT and 2A4 myoblasts and myotubes on days 0–4 of differentiation. The cell cycle inhibitor p21 was present in the nuclei of WT myotubes, starting on day 2 and persisting through day 4, but was not detectable in the nuclei of 2A4 cells at any point.

Despite the presence of MyoD and phosphorylated p53 in 2A4 myoblasts, p21 was not expressed, which is consistent with the observation that 2A4 cells continued to proliferate in differentiation medium depleted of mitogens. We hypothesized that MyoD, although present in the nucleus of 2A4 cells, is in state that prevents it from transcriptionally activating p21 and myogenin. To test this hypothesis, we examined the RNA levels of MyoD, p21, myogenin, and two negative regulators of myogenesis, Id2 and Id3 (inhibitors of differentiation 2 and 3). Id proteins antagonize basic helix-loop-helix (bHLH) transcription factors such as MyoD, and sequester them in an inactive conformation [59]. The Id genes are under the control of the transcription factor Egr-1, which is negatively regulated by the CaM-calcineurin complex during myogenesis [60, 61]. Indeed, we found that by day 4 of differentiation, WT myoblasts upregulated myogenin by 4-fold, p21 by 14-fold, and Rb by 7-fold; 2A4 myoblasts failed to upregulate myogenin, p21, or Rb (Fig 3D). MyoD expression level was only slightly upregulated with differentiation for both WT and 2A4 myoblasts, which contrasts immunofluorescence results (Fig 3), which indicates that MyoD protein disappears from WT myotubes on day 4, suggesting that MyoD protein is degraded by day 4 of differentiation in WT myotubes, but not in 2A4 myoblasts. As expected, the Id genes were both downregulated (2-fold) in WT myoblasts undergoing differentiation, which would relieve myogenesis suppression. In 2A4 myoblasts, however, the Id genes were upregulated 2-fold, which may indicate that the Id proteins remained available to antagonize MyoD, halting the differentiation program.

## CALM1 M109Q results in a dominant negative expression pattern

In humans, identical CaM proteins are encoded by three genes including *CALM1*, *CALM2*, and *CALM3*, located at distinct loci on three different chromosomes [62]. CaM is remarkably

conserved among vertebrates, and there are very few evolutionarily tolerated mutations [63]. The three CaM genes are not redundant in the sense that they ensure fine-tuning of CaM expression according to cell type; even slight dysregulation leads to aberrant physiology [62].

We found that the M109Q mutation in one or both alleles of *CALM1* leads to complete inhibition of myogenesis. How this dominant negative pattern of expression arises is puzzling, but is characteristic of CaM mutations. From 2012 to 2018, twenty-six pathogenic CaM mutations were described in the literature [63]; all CaM mutations with known pathology cause severe cardiac dysfunction, nearly all occur within $Ca^{2+}$ coordination sites in the C-terminal lobe, and all result in a dominant phenotype even though only 1/6 of the CaM pool contains the mutation [24]. As with the M109Q mutation, most of these mutations impact RyR activation and inhibition [64], leading to dysregulated $Ca^{2+}$ handling. No CaM mutations have been linked to pathological consequences apart from cardiac manifestations; with large-scale human genome sequencing on the horizon [65], it will be possible to discern broader consequences of human CaM mutations.

RyR mutations, too, give rise to cardiac dysfunction with a dominant phenotype. Because the RyR complex assembles as a tetramer, a single RyR mutation will result in nearly every complex containing at least one mutated RyR monomer, which destabilizes the entire complex, resulting in $Ca^{2+}$ leakage and ultimately, arrythmia [24]. A similar argument could be made for CaM: since many CaM targets (such as RyR) are multimers, a single mutant CaM could conceivably destabilize an entire multimeric complex. For example, CaMKII is a dodecameric complex that binds CaCaM in a highly cooperative manner, leading to sustained CaMKII activation through autophosphorylation; the cooperativity of CaCaM binding fine-tunes the frequency-dependent response of CaMKII activity to cellular $Ca^{2+}$ oscillations [66, 67]. In a dodecameric complex, if 1/6 of the CaM pool were mutated, on average, each dodecameric CaMKII complex would associate with two mutant CaMs, which could strongly alter cooperativity. We hypothesize that the *CALM1* M109Q mutation produces a dominant phenotype because of its association with multimeric complexes.

Direct in-cell measurements of RyR and CaMKII activity are not yet experimentally accessible, so right now we are not able to directly measure the impact of *CALM1*-M109Q on cellular RyR and CaMKII activation. Biochemical assays of RyR function in SR vesicles based on [³H] ryanodine binding indicate that CaM-M109Q cannot activate the RyR at low $Ca^{2+}$ and oxidized CaM cannot activate purified CaMKII at high $Ca^{2+}$ [32, 64]. Of course, it is imperative to understand RyR and CaMKII activation by CaM within the context of a muscle cell. The development of in-cell biochemical and biophysical measurements is therefore a priority, and progress has already been made in developing FRET-based assays for CaMKII within cells and RyR within SR vesicles [68, 69].

## Redox sensitive CaM pathways

$Ca^{2+}$ is a crucial component of the muscle differentiation medium [46], and $Ca^{2+}$ signaling pathways are now implicated at every step of myogenesis [47]. Downstream effectors of $Ca^{2+}$ are numerous, but CaCaM, CaCaM-dependent kinases, and CaCaM-calcineurin complexes play key roles in facilitating myogenesis [48]. Many of the kinases, phosphatases, pumps, channels, and regulatory proteins within these pathways have biochemically well-characterized redox sensitivities [54, 70–72], but nobody has reported the cellular impact of oxidizing a specific protein residue within one of these pathways until now. By using (1) a biochemically well-characterized mutation for mimicking methionine oxidation (Met to Gln), and (2) the CRISPR-Cas9 system to introduce point mutations directly into the genome, we were able to find that CaM acts as a potent cellular redox sensor that regulates myogenesis (Fig 4). We

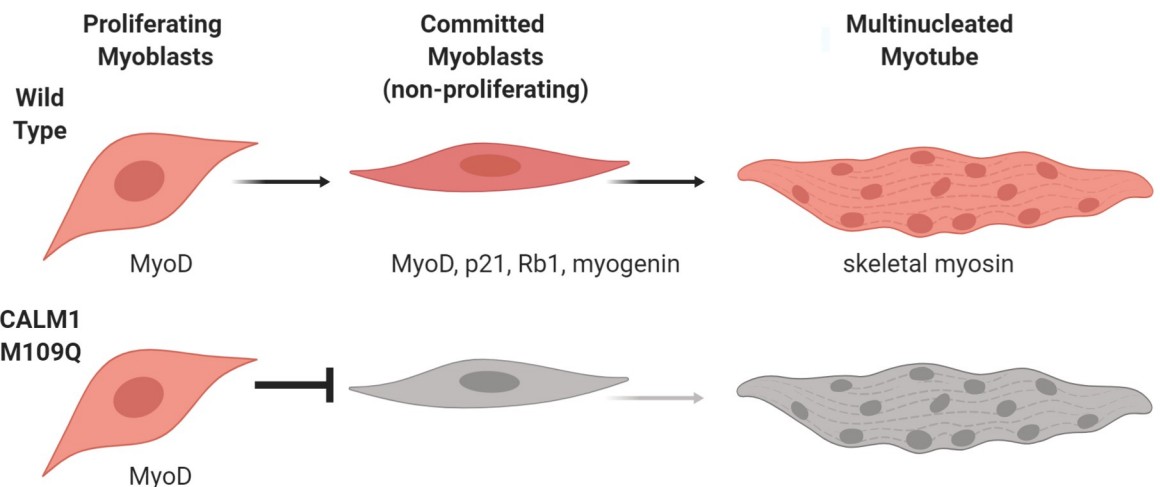

**Fig 4. CaM acts as a potent cellular redox sensor that regulates myogenesis.** The *CALM1* M109Q mutation, which mimics methionine oxidation at M109, blocks myogenesis before myoblasts exit from the cell cycle and commit to differentiation.

don't yet understand the precise mechanism through which CaM exerts its regulatory role, but we hypothesize that CaM's role as a redox sensor is coupled to its regulatory roles in the CaM-KII and calcineurin pathways.

Calcineurin contributes critically to the activity of the transcription factors MEF2, MyoD, and NFAT, which together drive transcription of myogenin [48, 60]. Calcineurin activates MEF2 via dephosphorylation in a signalosome complex localized to the nuclear envelope [73] and activates MyoD by decreasing the expression of the myogenic inhibitory factors Id1 and Id3 via transcriptional inactivation of Egr-1 [60, 61]. Here, we found that the Id genes are both downregulated 2-fold in WT myoblasts undergoing differentiation, but not in *CALM1* M109Q myoblasts, indicating that suppression of calcineurin activity might contribute to the myogenesis defect in mutant myoblasts. It isn't known whether CaM M109Q can bind or activate calcineurin, but biophysical data suggests that CaM M109Q doesn't populate the "active" $Ca^{2+}$-bound structure that activates most target proteins [38].

CaM-dependent kinase II (CaMKII) activity is required for the release of MEF2 from class II histone deacetylases, which bind MEF2 in an inactive complex [74]. Phosphorylation of the HDAC4/HDAC5 heterodimer by CaMKII directs their nuclear export [75], releasing MEF2, and relieving transcriptional repression of myogenin. An HDAC5 mutant lacking CaMK phosphorylation sites is resistant to CaMK-mediated nuclear export and acts as a dominant inhibitor of myogenesis [74]. It is conceivable that CaM M109Q blocks the multimeric CaM-KII complex from facilitating HDAC4/HDAC5 nuclear export, which would prevent MEF2 from activating myogenin expression and result in dominant inhibition of myogenesis.

Since CaM is a regulatory node in multiple $Ca^{2+}$ sensitive pathways, it is possible that the *CALM1* M109Q mutation exerts a strong influence along multiple coordinated paths, ultimately blocking myogenesis. CaM Met oxidation is reversible by the Met sulfoxide reductase enzymes [76] so that when the cellular redox environment is restored to normal, the inhibitory effect of CaM Met oxidation is relieved. With chronic oxidative stress, as occurs with age-related degenerative disease, CaM's regulatory influence is likely to contribute to blunted muscle regeneration, weakness, and morbidity. We have shown for the first time that a single oxidative modification to a redox-sensitive muscle regulatory protein can halt myogenesis, suggesting a molecular target for mitigating the impact of oxidative stress in age-related muscle degeneration.

## Limitations of the study

C2C12 myoblasts are a suitable model system for exploring early hypotheses relating site-specific muscle protein oxidation and muscle function, but they can't entirely recapitulate the function of muscle satellite cells. While a mouse model would clarify the impact of muscle protein oxidation on muscle aging, freshly isolated muscle satellite cells or human induced pluripotent stem cells (iPSCs) could be edited and re-grafted onto human muscle to more directly understand the impact of muscle protein oxidation (or blocking oxidation) on human muscle physiology [77]. Indeed, the current work establishes the rational for using gene-editing to create better models of age-related degenerative disease and for using gene-editing to alleviate these conditions.

## Supporting information

**S1 Raw images. Western blots for Fig 3C.**
(PDF)

**S1 Dataset. Differentiation assays.** Percentage of nuclei expressing myogenin and percentage of nuclei in myotubes for wild type (WT) and *CALM1* M109Q (2A4, 3C2, and 3C3) C2C12 cells treated with hydrogen peroxide treatment and various metabolic inhibitors.
(XLSX)

**S2 Dataset. Cell proliferation assays.** Cell proliferation measured by the MTS assay for wild type (WT) and *CALM1* M109Q (2A4 and 3C2) C2C12 cells in growth or differentiation medium and with hydrogen peroxide treatment.
(XLSX)

**S3 Dataset. Immunofluorescence images.** Representative immunofluorescence images acquired for various myogenic factors and muscle proteins on days 0–4 of differentiation for wild type (WT) and *CALM1* M109Q (2A4) C2C12 cells.
(PDF)

**S4 Dataset. MyoD fluorescence intensity.** Intensity of nuclear MyoD fluorescence in wild type (WT) and *CALM1* M109Q (2A4) C2C12 cells on days 0–4 of differentiation.
(XLSX)

**S5 Dataset. Quantitative PCR.** $C_q$ values for various myogenic factors and muscle proteins in wild type (WT) and *CALM1* M109Q (2A4) cells on day 0 and day 4 of differentiation.
(XLSX)

## Acknowledgments

Students at the University of Wisconsin-La Crosse in Molecular Biology (Bio 436/536, Spring 2017, Spring 2018 and Fall 2019) and Advanced Microscopy (Bio 449/549, Fall 2018) taught by Dr. Jennifer Klein and Dr. Scott Cooper contributed to assay development and preliminary results.

## Author Contributions

**Conceptualization:** Alex W. Steil, Jennifer C. Klein.

**Data curation:** Alex W. Steil, Jacob W. Kailing, Cade J. Armstrong, Daniel G. Walgenbach.

**Formal analysis:** Jacob W. Kailing, Jennifer C. Klein.

**Writing – original draft:** Jennifer C. Klein.

**Writing – review & editing:** Jennifer C. Klein.

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
