## [Decision Letter · Decision Letter 0]

18 Feb 2020

PONE-D-20-00434

The calmodulin redox sensor controls myogenesis

PLOS ONE

Dear Dr. Klein,

Thank you for submitting your manuscript to PLOS ONE. After careful consideration, we feel that it has merit but does not fully meet PLOS ONE’s publication criteria as it currently stands. Therefore, we invite you to submit a revised version of the manuscript that addresses the points raised during the review process.

We would appreciate receiving your revised manuscript by Apr 03 2020 11:59PM. To enhance the reproducibility of your results, we recommend that if applicable you deposit your laboratory protocols in protocols.io, where a protocol can be assigned its own identifier (DOI) such that it can be cited independently in the future. For instructions see: http://journals.plos.org/plosone/s/submission-guidelines#loc-laboratory-protocols

We look forward to receiving your revised manuscript.

Kind regards,

Atsushi Asakura, Ph.D

Academic Editor

PLOS ONE

Journal Requirements:

Reviewers' comments:

Reviewer's Responses to Questions

**Comments to the Author**

1. Is the manuscript technically sound, and do the data support the conclusions?

Reviewer #1: Partly

Reviewer #2: No

2. Has the statistical analysis been performed appropriately and rigorously? 

Reviewer #1: No

Reviewer #2: Yes

3. Have the authors made all data underlying the findings in their manuscript fully available?

Reviewer #1: Yes

Reviewer #2: No

4. Is the manuscript presented in an intelligible fashion and written in standard English?

Reviewer #1: Yes

Reviewer #2: No

5. Review Comments to the Author

Reviewer #1: The study aimed to address the question of the role of a redox-sensitive CAM1 in aging muscle. The CRISPR-Cas9 method was used to generate a CAM1-M109Q mutant that mimicked the oxidative modification of one of 9 methonines in one or both alleles of the three expressed CaM genes in C2C12 mouse myoblast. A potentially important conclusion was that the modification of a single methione was sufficient to halt myogenesis in aging muscle.

Using a skeletal muscle mouse line, the authors investigated the functional effects of a CAM mutant that mimics the oxidation of a methionine to methionine sulfoxide. By using immunofluorescence, cell proliferation and RT-PCR assays, they then showed that the mutant inhibited the expression of late myogenic factors in C2C12 cells. Overall, the cellular studies seem to have been well done, however, they are limited in scope. The study is largely correlative and is limited to one of the multiple mechanisms under the cellular control of CaM. As stated by the authors, methionine oxidation impairs the regulation of numerous target proteins. CaM mutants including the CaM-M109Q have been shown to disrupt the function of the ryanodine receptor and it is surprising that this was not addressed. Performing these experiments is strongly recommended.

Second, a potential experimental limitation of CRISPR-Cas9 method is that secondary unwanted mutations are introduced, leading to the observed disruptive events. I miss a clear description of why this did not occur in the present study.

Fig. 1B, show also the structure of CaM-M109Q

Fig. 2 C and D and Fig. 3B , number of independent experiments should be shown.

Fig. 2D in text?

Reviewer #2: Manuscript Number: PONE-D-20-00434

“The calmodulin redox sensor controls myogenesis” by Klein J. et al.

Skeletal muscle regeneration is largely dependent on resident muscle stem cells that are quickly activated and differentiated in response to muscle injury. However, the capacity of skeletal muscle to regenerate declines with aging. Oxidative stress is one of the central mechanisms for the blunted muscle regeneration with aging. In this manuscript, the authors focus on one of muscle redox sensors, calmodulin in C2C12 mouse myoblasts. The authors use the CRISPR-Cas9 system to induce a single amino acid substitution of calmodulin that mimics oxidation of methionine in calmodulin. Mutated calmodulin in C2C12 myoblasts negatively affects the myogenic differentiation with low late myogenic markers such as myogenin and myosin heavy chain. This is an interesting concept that describes the single amino acid substitution of calmodulin inhibits the differentiation of C2C12 myoblasts. However, there are some major considerations that should be addressed.

Essential revisions:

1. The authors describe that 2A4 line contains the mutation on both alleles, on the other hand, 3C2 and 3C3 lines contain the desired mutation in only one allele. In the manuscript, the authors have concluded that all three lines are identical phenotypes. However, data about all three line is only Figure 2C (IF for myogenin). The authors should include at least one of cell lines with one allele mutation (3C2 or 3C3) in revised version of Figure 2D and Figure 3A, B.

2. Is there any evidence that mutated calmodulin in C2C12 myoblasts cannot fully function? The authors should show the data about the point, for example, the activation of CaMKII between WT vs mutant myoblasts, etc.

3. According to Figure 3A, it seems that MyoD signals in 2A4 cells are much stronger than those in WT at day 4, which make sense to me if the mutant myoblasts fail to withdraw from the cell cycle and keep proliferating (Figure 2D). However, mRNA expression of MyoD in 2A4 cells is lower than WT in Figure 3B. The contradictory result should be clarified by performing western blot (WB) analysis. In addition to this, results of Figure 3A (differentiation defect with low late myogenic factors) would be strengthened by WB, at least for Myogenin and MyHC.

4. I’m not in favor of reporting q-PCR in Figure 3B by normalizing to the day 0 condition. It is better to compare the expression of each gene between WT vs 2A4 at day 4. I recommend that the authors include either 3C2 or 3C3 lines containing the one allele mutation in the revised figure as I mentioned above.

5. H2O2 treatment inhibits the myogenic differentiation according to Figure 2B, does this condition induce the oxidation of methionine in calmodulin? The author should clarify this point.

6. The mutated myoblasts proliferate faster than WT myoblast in both GM and DM conditions as shown in Figure 2D. Does H2O2 treatment also increase the proliferation of C2C12 myoblasts?

7. Which results support the Figure 4A and 4B? I think none of these data described in this manuscript support Figure 4A and 4B. The data described in this paper could only lead a conclusion that the single amino acid mutation in muscle redox sensor calmodulin fails to withdraw the cell cycle and directly/indirectly suppresses the late myogenic factors such as myogenin. If the authors want to support the conclusion as shown in Figure 4A-B, the authors should perform to investigate the transcriptional activity of these genes by luciferase assay in the proper condition. I recommend that the authors delete and replace it with another graphical summary which explains what you found in this study.

Minor points:

8. Page 11, “The calmodulin redox sensor halts myogenesis”, the authors should check the numerical order of Figure 2. There are a lot of mislabeling.

6. PLOS authors have the option to publish the peer review history of their article (what does this mean?). If published, this will include your full peer review and any attached files.

Reviewer #1: No

Reviewer #2: No

---

## [Author Response · Author response to Decision Letter 0]

15 Aug 2020

August 7, 2020

Dear Editor, 

The reviewer comments were extremely helpful in creating a manuscript that more clearly describes our results and more carefully contextualizes them. We apologize for the delay as our research lab has been closed since the second week of March due to COVID and has not yet reopened, nor have any students returned to campus. Our response to each reviewer comment is below. The reviewer comment/question is followed by our response and a description of changes we made to the manuscript. Because of some significant additions to our figures, we have had to make additions to the methods section. All changes within the actual manuscript have been highlighted in yellow and changes have been tracked.

Reviewer #1: 

• The study is largely correlative and is limited to one of the multiple mechanisms under the cellular control of CaM. As stated by the authors, methionine oxidation impairs the regulation of numerous target proteins. 

Since oxidized CaM has been recovered from aged tissue, our earliest investigations were focused on cell mechanisms known to be altered by aging. We added these results as Figure 2F that includes an investigation of whether the CALM1-M109Q mutation alters mitochondrial structure or lysosome distribution and dynamics, both of which are redox regulated cell processes involving CaM. We didn’t observe any impact due to CALM1-M109Q. In addition to Figure 2F, we added the following text to a new (second) paragraph of the results section entitled The calmodulin redox sensor halts myogenesis:

We investigated mitochondrial structure and lysosome distribution in CALM1 M109Q myoblasts with live-cell imaging using MitoTracker and LyosTracker to detect redox-regulated alterations to metabolism or autophagy, but didn’t observe any differences between WT and 2A4 cells (Fig. 2F), even with glucose starvation (not shown).

Early on, we hypothesized that the myogenesis defect was due to over-activation of the AMPK nutrient-sensing pathway, which involves multiple CaM-dependent targets and Ca2+ release from the RyR, a pathway that is dysregulated with aging. We attempted to rescue the myogenesis defect using inhibitors and activators of the AMPK pathway including dorsomorphin (AMPK inhibitor), STO-609 (CaMKK inhibitor), and EX-528 (Sirt1 inhibitor), but none of these interventions recovered myogenesis. In addition to Figure 2G, we added the following text to a new (second) paragraph of the results section entitled The calmodulin redox sensor halts myogenesis:

The AMPK nutrient-sensing pathway, which involves Ca2+ release and multiple CaM-dependent targets, negatively regulates myogenesis (Fulco et al., 2008; Williamson et al., 2009), so we hypothesized that inhibitors of this pathway might recover myogenesis in CALM1-M109Q myoblasts. We found that although the AMPK inhibitor dorsomorphin and the CaMKKβ inhibitor STO-609 both boosted myogenesis in WT myoblasts, they failed to recover myogenesis in 2A4 cells (Fig. 2G).

We embarked on this study with an interest in characterizing Ca2+ dynamics in myotubes, as the Ca2+ transient is the complex result of the coordination of the pumps, channels, and kinases that CaM regulates. Regrettably, the CaM-M109Q mutation prevented the formation of myotubes. Myoblasts don’t produce Ca2+ transients because they don’t express the machinery for it. Therefore, a future study will include measuring the impact of CaM-M109Q on Ca2+ dynamics in myotubes, perhaps using a regulated promoter so that CaM-M109Q expression can be turned on after myogenesis.

• CaM mutants including the CaM-M109Q have been shown to disrupt the function of the ryanodine receptor and it is surprising that this was not addressed. Performing these experiments is strongly recommended.

We’ve clarified the state of in-cell protein function assays for CaM-dependent channels and proteins so that it is clear what has been done already and what is not yet possible in a new, fourth paragraph in the section entitled “CALM1 M109Q results in a dominant negative expression pattern:”

Direct in-cell measurements of RyR and CaMKII activity are not yet experimentally accessible, so right now we are not able to directly measure the impact of CALM1-M109Q on cellular RyR and CaMKII activation. Biochemical assays of RyR function in SR vesicles based on 3H ryanodine binding indicate that CaM-M109Q cannot activate the RyR at low Ca2+ and oxidized CaM cannot activate purified CaMKII at high Ca2+ (Balog et al., 2003; Robison et al., 2007). Of course, it is imperative to understand RyR and CaMKII activation by CaM within the context of a muscle cell. The development of in-cell biochemical and biophysical measurements is therefore a priority, and progress has already been made in developing FRET-based assays for CaMKII within cells and RyR within SR vesicles (Ardestani et al., 2019; Rebbeck et al., 2017). 

Collaborators are working to develop FRET-based assays for both RyR and CaMKII using a new, high-throughput transient time-resolved fluorescence instrument that we hope to apply to this project very soon. The ability to make these in-cell functional measurements has been directly coupled to recent innovations in fluorescence detection.

• Second, a potential experimental limitation of CRISPR-Cas9 method is that secondary unwanted mutations are introduced, leading to the observed disruptive events. I miss a clear description of why this did not occur in the present study.

We clarified our reasoning for not pursuing whole genome sequencing by adding the following to the second paragraph of the section entitled CALM1 gene editing in mouse myoblasts:

We did not pursue whole genome sequencing to confirm the absence of unwanted mutations because there were no predicted off-target cutting sites in the mouse genome and the probability that three independent cell lines would contain both the desired mutation and the same off-target mutation would be < 0.01% given an optimistic cutting efficiency of 20%. 

Also, whole genome sequencing easily misses large-scale chromosomal deletions, insertions, and rearrangements that could occur near the on-target region, but these alterations would have resulted in the loss of a primer binding site or would produce an amplicon too large to be amplified, so our amplicon-based selection procedure would have selected against cells with these alterations.

• Fig. 1B, show also the structure of CaM-M109Q

Unfortunately, there are no crystal structures for CaM-M109Q. Other biophysical methods such as circular dichroism suggest that CaM secondary and tertiary structure do not change as a result of the M109Q mutation. We published a computational study that indicated that the M109Q mutation doesn’t grossly alter CaM structure, but does strongly alter the conformational transition and equilibrium between CaM’s open and closed states. We’ve clarified this point in the second paragraph of the introduction to now read as follows:

Our group has delineated the molecular structural mechanism by which oxidation of CaM’s C-terminal Met (M109 and M124, see Fig. 1B) trigger changes in CaM conformational dynamics without changing the secondary or tertiary structure of CaM (McCarthy et al., 2015; Walgenbach et al., 2018).

• Fig. 2C and D and Fig. 3B , number of independent experiments should be shown.

The number of independent experiments is now described in the legend for Figure 2 and Figure 3 as follows: Fig. 2B, n=4; Fig. 2C, n=4, Fig. 2D, n=2; and Fig. 3B, n=3.

• Fig. 2D in text?

References to figures in the text were corrected.

Reviewer #2

Essential revisions:

• The authors describe that 2A4 line contains the mutation on both alleles, on the other hand, 3C2 and 3C3 lines contain the desired mutation in only one allele. In the manuscript, the authors have concluded that all three lines are identical phenotypes. However, data about all three line is only Figure 2C (IF for myogenin). The authors should include at least one of cell lines with one allele mutation (3C2 or 3C3) in revised version of Figure 2D and Figure 3A, B.

We have now included data for cell line 3C2 in Fig. 2D. With the continued closure of my research lab due to COVID, we are not able to acquire new data for Fig. 3. 

• Is there any evidence that mutated calmodulin in C2C12 myoblasts cannot fully function? The authors should show the data about the point, for example, the activation of CaMKII between WT vs mutant myoblasts, etc.

We added panels Fig. 2F and Fig. 2G that include investigations of whether the CALM1-M109Q mutation alters mitochondrial structure or lysosome distribution, and whether the myogenesis defect could be rescued by inhibiting the AMPK pathway, as described above.

• According to Figure 3A, it seems that MyoD signals in 2A4 cells are much stronger than those in WT at day 4, which make sense to me if the mutant myoblasts fail to withdraw from the cell cycle and keep proliferating (Figure 2D). However, mRNA expression of MyoD in 2A4 cells is lower than WT in Figure 3B. The contradictory result should be clarified by performing western blot (WB) analysis. In addition to this, results of Figure 3A (differentiation defect with low late myogenic factors) would be strengthened by WB, at least for Myogenin and MyHC.

We have now included western blots for MyoD in what is now Fig. 3C. We also quantified the fluorescence intensity due to MyoD in nuclei in Fig. 3B. Indeed, we did find that MyoD protein levels are higher in 2A4 cells than WT cells on day 4 of differentiation.

We added the following text to the results section entitled The calmodulin redox sensor halts myogenesis:

Paragraph three:

Quantification of fluorescence intensity of MyoD in nuclei (Fig. 3B) and western blotting (Fig. 3C) indicated that MyoD protein disappeared from WT nuclei on day 4, consistent with its degradation by the 26S proteasome (Abu Hatoum et al., 1998), while MyoD protein increased by day 4 in 2A4 nuclei.

Paragraph five:

MyoD expression level was unchanged with differentiation for both WT and 2A4 myoblasts, which contrasts immunofluorescence results that indicate MyoD protein is degraded by day 4 of differentiation in WT myotubes, but not in 2A4 myoblasts.

• I’m not in favor of reporting q-PCR in Figure 3B by normalizing to the day 0 condition. It is better to compare the expression of each gene between WT vs 2A4 at day 4. I recommend that the authors include either 3C2 or 3C3 lines containing the one allele mutation in the revised figure as I mentioned above.

Fig. 3B is now Fig. 3D and has been changed so that the fold change between WT and 2A4 at day 4 is presented, in addition to the original presentation, as we found both to be helpful.

• H2O2 treatment inhibits the myogenic differentiation according to Figure 2B, does this condition induce the oxidation of methionine in calmodulin? The author should clarify this point.

We have not quantified the extent of calmodulin methionine oxidation with H2O2 treatment as this requires sophisticated mass spectrometry techniques inaccessible to us. However, collaborators have estimated the extent of CaM and RyR oxidation by H2O2 and the resulting functional decline in myocytes, so we’ve included this information in the results section entitled The calmodulin redox sensor halts myogenesis:

We treated myoblasts with H2O2 concentrations over which CaM and other targets are expected to be significantly oxidized. When rat ventricular myocytes were exposed to 50 µM H2O2, there was a 50% reduction in free thiols on RyR and a 50% decrease in CaM binding to RyR, an effect attributed to oxidation of both RyR and CaM; pre-treating CaM with 50 µM H2O2 before perfusion into myocytes (without additional H2O2-treatment) produced a 20% decline in RyR binding (Oda et al., 2015). 

• The mutated myoblasts proliferate faster than WT myoblast in both GM and DM conditions as shown in Figure 2D. Does H2O2 treatment also increase the proliferation of C2C12 myoblasts?

We have included panel Fig. 2E that includes cell proliferation data for WT H2O2-treated C2C12 cells and have included a response to the reviewer question in the first paragraph of the results section entitled The calmodulin redox sensor halts myogenesis:

We found that 25 �M H2O2 enhanced cell proliferation for intermediate time points, but as the H2O2 concentration approached 100 µM, cells stopped proliferating entirely (Fig. 2E). It has been observed that H2O2 can enhance proliferation at low concentrations, arrest growth at high concentrations, and elicit apoptosis at even higher concentrations (Giorgio et al., 2007).

• Which results support the Figure 4A and 4B? I think none of these data described in this manuscript support Figure 4A and 4B. The data described in this paper could only lead a conclusion that the single amino acid mutation in muscle redox sensor calmodulin fails to withdraw the cell cycle and directly/indirectly suppresses the late myogenic factors such as myogenin. If the authors want to support the conclusion as shown in Figure 4A-B, the authors should perform to investigate the transcriptional activity of these genes by luciferase assay in the proper condition. I recommend that the authors delete and replace it with another graphical summary which explains what you found in this study.

We have replaced Figure 4A and 4B with a graphical summary that doesn’t overstate our conclusion.

• Minor points:

8. Page 11, “The calmodulin redox sensor halts myogenesis”, the authors should check the numerical order of Figure 2. There are a lot of mislabeling.

References to figures in the text have been corrected.

Thank you for your consideration of this much improved manuscript.

Sincerely, 

Jennifer Klein

---

## [Decision Letter · Decision Letter 1]

31 Aug 2020

The calmodulin redox sensor controls myogenesis

PONE-D-20-00434R1

Dear Dr. Klein,

We’re pleased to inform you that your manuscript has been judged scientifically suitable for publication and will be formally accepted for publication once it meets all outstanding technical requirements.

Kind regards,

Atsushi Asakura, Ph.D

Academic Editor

PLOS ONE

Additional Editor Comments (optional):

Reviewers' comments:

Reviewer's Responses to Questions

**Comments to the Author**

1. If the authors have adequately addressed your comments raised in a previous round of review and you feel that this manuscript is now acceptable for publication, you may indicate that here to bypass the “Comments to the Author” section, enter your conflict of interest statement in the “Confidential to Editor” section, and submit your "Accept" recommendation.

Reviewer #1: All comments have been addressed

Reviewer #2: All comments have been addressed

2. Is the manuscript technically sound, and do the data support the conclusions?

Reviewer #1: Yes

Reviewer #2: Yes

3. Has the statistical analysis been performed appropriately and rigorously? 

Reviewer #1: Yes

Reviewer #2: Yes

4. Have the authors made all data underlying the findings in their manuscript fully available?

Reviewer #1: Yes

Reviewer #2: Yes

5. Is the manuscript presented in an intelligible fashion and written in standard English?

Reviewer #1: Yes

Reviewer #2: Yes

6. Review Comments to the Author

Reviewer #1: There are no comments for the authors. There are no comments for the auithors. There are no comments for the authors.

how many characters are needed?

Reviewer #2: I have read with interest the revised version of the manuscript, which has been improved by the additional requested evidence. I therefore recommend publication on PLOS ONE.

7. PLOS authors have the option to publish the peer review history of their article (what does this mean?). If published, this will include your full peer review and any attached files.

Reviewer #1: No

Reviewer #2: No

---

## [Editor Report · Acceptance letter]

8 Sep 2020

PONE-D-20-00434R1 

The calmodulin redox sensor controls myogenesis 

Dear Dr. Klein:

I'm pleased to inform you that your manuscript has been deemed suitable for publication in PLOS ONE. Congratulations! Your manuscript is now with our production department. 

Kind regards, 

on behalf of

Dr. Atsushi Asakura 

Academic Editor

PLOS ONE